# Engineered Fragments of the PSMA-Specific 5D3 Antibody and Their Functional Characterization

**DOI:** 10.3390/ijms21186672

**Published:** 2020-09-12

**Authors:** Zora Novakova, Nikola Belousova, Catherine A. Foss, Barbora Havlinova, Marketa Gresova, Gargi Das, Ala Lisok, Adam Prada, Marketa Barinkova, Martin Hubalek, Martin G. Pomper, Cyril Barinka

**Affiliations:** 1Laboratory of Structural Biology, Institute of Biotechnology of the Czech Academy of Sciences, BIOCEV, Prumyslova 595, 252 50 Vestec, Czech Republic; nikolabelousova@gmail.com (N.B.); barbora.havlinova@ibt.cas.cz (B.H.); marketa.gresova@ibt.cas.cz (M.G.); gargi.das@ibt.cas.cz (G.D.); adamprada@gmail.com (A.P.); marketabarinkova@gmail.com (M.B.); 2The Russell H. Morgan Department of Radiology and Radiological Science, Johns Hopkins Medical Institutions, 1550 Orleans St, Baltimore, MD 21231, USA; cfoss1@jhmi.edu (C.A.F.); alisok1@jhmi.edu (A.L.); mpomper@jhmi.edu (M.G.P.); 3Mass Spectrometry Group, Institute of Organic Chemistry and Biochemistry of the Czech Academy of Sciences, IOCB & Gilead Research Center, Flemingovo náměstí 542/2, 160 00 Prague, Czech Republic; martin.hubalek@uochb.cas.cz

**Keywords:** prostate-specific membrane antigen, in vivo imaging, prostate cancer, monoclonal antibody, antibody fragment, glutamate carboxypeptidase II, NAALADase

## Abstract

Prostate-Specific Membrane Antigen (PSMA) is an established biomarker for the imaging and experimental therapy of prostate cancer (PCa), as it is strongly upregulated in high-grade primary, androgen-independent, and metastatic lesions. Here, we report on the development and functional characterization of recombinant single-chain Fv (scFv) and Fab fragments derived from the 5D3 PSMA-specific monoclonal antibody (mAb). These fragments were engineered, heterologously expressed in insect S2 cells, and purified to homogeneity with yields up to 20 mg/L. In vitro assays including ELISA, immunofluorescence and flow cytometry, revealed that the fragments retain the nanomolar affinity and single target specificity of the parent 5D3 antibody. Importantly, using a murine xenograft model of PCa, we verified the suitability of fluorescently labeled fragments for in vivo imaging of PSMA-positive tumors and compared their pharmacokinetics and tissue distribution to the parent mAb. Collectively, our data provide an experimental basis for the further development of 5D3 recombinant fragments for future clinical use.

## 1. Introduction

The increasing length of the human lifespan brings about higher incidence of various health problems including cancer. Prostate cancer (PCa) holds a prominent position among tumor-associated diseases among elderly men in industrial countries due to its high incidence and associated mortality rates of metastatic disease [1,2,3]. To improve PCa prognosis, a combination of precise diagnosis of the tumor stage and specifically designed treatment plans is required. In the clinic today, PCa is routinely detected using imaging techniques such as bone scintigraphy, computed tomography, ultrasound, and magnetic resonance imaging (reviewed in [4]). However, there is still an untapped potential in PCa diagnostics and staging in terms of sensitivity and specificity. Thus, new PCa-specific reagents are extensively researched.

Antibody fragments are considered to be an excellent platform being developed for cancer-related diagnostic and therapeutic approaches, particularly for targeted therapy (reviewed by [5]). Single-chain variable fragments (scFv), Fab fragments and nanobodies are the primary medicinally relevant antibody fragments. They may serve as antigen-targeting moieties in various conjugates, fusion proteins, and bispecific molecules. In contrast to whole immunoglobulin complexes, fragments feature enhanced tumor penetration, fast blood clearance, optimal half-life and biodistribution, low immunogenicity, and low imaging background [6]. Due to these qualities, fragments promote the uptake of drugs by tumor cells, increase the specificity of anti-tumor drug delivery and reduce undesirable cytotoxic side effects in healthy tissues. Furthermore, they can trigger anti-tumor immune responses and collectively modulate the function of immune cells, as well as inhibit the proliferation of tumorigenic cells [7,8,9,10]. Coupled with radionuclides, fluorescent dyes, or nanoparticles, antibody fragments serve as imaging modalities for noninvasive detection and staging of tumors, and the evaluation of the surface molecule portfolio of cancer cells. Moreover, fragment-based imaging provides additional information beneficial for treatment including real-time fluorescence-guided surgery [11,12,13,14].

Prostate-specific membrane antigen (PSMA) is one of the leading PCa-specific biomarkers strongly expressed on PCa cells [15]. The specific presence of PSMA in primary, high-grade, androgen-independent, and metastatic PCa tumors predetermines the enzyme as a prime tool for PCa imaging and therapy [16,17,18,19]. Consequently, PSMA-specific antibodies are being developed as new PCa-targeting theranostics. Various constructs have already been tested in vitro on PSMA-expressing cells [20] as well as in vivo in relevant xenograft models [16,21,22,23,24,25]. Notably, PCa imaging in human patients has also been performed using a PSMA-specific ^89^Zr-IAB2M minibody [26].

Several PSMA-targeting entities are being developed as PCa therapeutics. Site-specific delivery of Notch1-specific siRNA by an anti-PSMA scFv was documented to inhibit PCa tumor growth [27]. Multimeric structures derived from anti-PSMA scFv and bispecific molecules, namely anti-PSMA/anti-CD3 fusions, have been used in preclinical immunotherapy models [28,29,30,31,32,33,34]. Immunotoxin fusions derived from PSMA-specific scFvs showed efficient control of PCa xenografts and enhanced anti-tumor activity of cytotoxic drugs [35,36,37,38]. Interestingly, PSMA-specific antibody fragments were documented to direct the cytotoxicity of chimeric antigen receptor T cells (CAR-T cells) not only to PCa but also to ovarian cancer [39,40]. In virotherapy approaches, oncolytic viruses targeted to cancer cells via anti-PSMA scFv revealed potential to induce PCa regression [41,42].

In this report, we exploit the superior features of our newly developed 5D3 mAb, including sub-nanomolar affinity and a high specificity for native PSMA [43]. These characteristics make the 5D3 mAb particularly suitable for in vivo applications [44,45,46]. Here, we cloned, heterologously expressed, purified, and characterized engineered scFv and Fab fragments of 5D3. Affinity and specificity of the recombinant fragments comparable to the parent antibody, along with the demonstrated applicability for in vivo imaging suggests that the binding site remained intact and the function of the recombinant proteins was retained throughout the engineering process. The study shows a high potential for the further development of 5D3-based engineered molecules, which could enter clinical trials for PCa imaging and therapy.

## 2. Results

### 2.1. Identification of Nucleotide and Amino Acid Sequences of the 5D3 mAb

Using N-terminal amino acid sequencing, we identified DIQMTQTNS and EVQPQVSKTMA peptides to be N-termini of light and heavy chains of the 5D3 mAb, respectively. This information was then used to design degenerated PCR primers to amplify sequences encoding the light and heavy chain of the 5D3 Fab fragment using a cDNA template isolated from 5D3 hybridomas. The primary amino acid sequences of 5D3 Fab, as derived from the amplified nucleotide sequences, were then matched against mass spectrometry data of peptides from tryptic and chymotryptic digests of purified full-length 5D3 mAb. Thanks to the complete sequence coverage and 100% match between the predicted and experimental primary amino acid sequences, we verified that the amplified genes really encode the 5D3 mAb that can be used for further engineering efforts.

### 2.2. Characterization of 5D3 Antibody Fragments Produced in E. coli

To assess whether 5D3 recombinant antibody fragments retained the specificity and high affinity of the original 5D3 antibody, we cloned Fab and scFv fragments into vectors used for heterologous protein expression in the *E. coli* periplasmic space. For Fab production, sequences encoding 5D3 variable domains were cloned into a pASK85 bicistronic expression plasmid that carries genes for constant domains of mouse Cκ and CH1γ1 subclasses (Figure 1) [47,48]. In the case of scFvs, their folding and function is affected by the order of variable domains, the length of the linker between domains, the type of a signal sequence, and the presence and position of purification tags. Consequently, we designed two scFv constructs differing in the order of variable domains, denoted LH and HL, the position of a 6xHis-tag and the sequence of a secretion signal (Figure 1). An identical 18-mer linker was inserted between the heavy and light chain domains in both variants and constructs cloned into the pASK85 vector backbone. All recombinant proteins were secreted into the *E. coli* periplasmic space and purified by a single step nickel chelating affinity chromatography using established protocols [49]. Large scale expressions yielded on average 20–30 µg protein/liter culture with an estimated purity of >90% (Figure 2A).

The specificity of purified antibody derivatives was evaluated by indirect immunofluorescence (IF) microscopy and flow cytometry using HEK293T cells overexpressing PSMA [43] and PCa-derived cell lines DU145 (PSMA-negative) and LNCaP (PSMA-positive). The flow cytometry analysis revealed a specific fluorescence signal on PSMA-positive LNCaP cells, while DU145 (PSMA-negative) cells were not labeled. Additionally, signal intensities for both scFv variants as well as the recombinant Fab fragment were comparable to staining with parent 5D3 mAb suggesting that all engineered fragments retained high affinity and specificity for PSMA (Figure 2B). In a complementary IF microscopy approach, a specific signal was also detectable only in samples of PSMA-positive cells—on the surface of non-permeabilized cells, and on the surface and in the cytoplasm of Triton-permeabilized cells (Figure 2C). It shall be noted that a weak non-specific signal was visible in the nuclear region in all samples permeabilized by Triton X-100. This signal is linked to cross-reactivity of the anti-6× His antibody (used as a detection reagent) towards polyhistidine sequences present in nuclear proteins [50].

### 2.3. Expression of Fab and scFv Fragments in Insect S2 Cells

As yields of 5D3 fragments expressed in *E. coli* were rather low and would not allow us to carry out their detailed in vitro/in vivo characterization, we directed the expression of the fragments to insect S2 cells. Genes encoding 5D3 fragments were recloned into a pMT expression vector in frame with a BiP secretion signal sequence and the SA-strep tag sequence was added to the C-terminus of the Fab heavy chain and scFv variants (Figure 1). In the case of Fab, expression vectors containing individual light and heavy Fab chains were co-transfected to Schneider’s S2 insect cells at a 2:1 ratio.

Engineered fragments were purified from the conditioned media by a combination of Streptactin affinity and size exclusion chromatography (SEC) and predominant monomeric protein species were observed during SEC runs as expected (Figure 3). Dimers of scFv variants represented <7% of the monomer peak (Figure 3B,C, fractions 3 and 4, respectively). All preparations were highly pure (>95%) with final yields of the Fab and scFv HL variants in the range of 10–20 mg/L culture, whereas the yield of the scFv LH variant was approximately 1 mg/L.

Surprisingly, SDS-PAGE analysis of purified fragments revealed an extra band of a higher molecular weight (app. 2 kDa) in all samples (Figure 3). Subsequent MS analysis identified slower migrating bands as fusions with the uncleaved signal peptide (Appendix A). Apparently, the S2 secretion system can be overwhelmed by robust production of recombinant proteins, and a fraction of secreted proteins is thus not processed properly. However, as shown in the following experiments, the presence of the uncleaved signal peptide does not have any negative influence on the performance of engineered fragments.

### 2.4. In Vitro Characterization of S2-Produced 5D3 Variants

In a preliminary set of experiments, we determined the stability, specificity, and affinity of S2-produced 5D3 fragments in vitro. Thermal stability of the engineered variants was determined by differential scanning fluorimetry (nanoDSF) and the results are shown in Figure 4A. scFv HL and LH variants unfolded at similar melting temperatures (Tm) of 53.4 °C and 52.5 °C, respectively, while the stability of the Fab fragment was higher as evidenced by Tm = 71.3 °C. The melting temperature of the engineered Fab fragment was similar to the Tm of the intact parent antibody (Tm = 70.7 °C) and the data are in line with results on the stability of mouse IgG1 published previously [51].

To evaluate the specificity of the variants, we used indirect immunofluorescence microscopy and flow cytometry in setups similar to those described above for variants produced in *E. coli*. The most notable technical difference was the use of an anti-Strep monoclonal antibody for the detection of the 5D3 fragments. As expected, the staining pattern in the IF microscopy experiment was virtually identical to the one performed on *E. coli*-expressed variants revealing specific plasma membrane/cytoplasmic staining in Triton-treated samples (Figure 4B). Likewise, only PSMA-positive (LNCaP, PC-3 PIP and CW22Rv1) cells were labeled in flow cytometry experiments, while the fluorescence signal for PSMA-negative cell lines (DU145 and PC-3) was close to background (Figure 4C and Appendix A).

Finally, the affinity of engineered molecules was evaluated by native ELISA and flow cytometry. For native ELISA, purified biotinylated PSMA was immobilized on streptavidin-coated plates and probed using a 2-fold dilution series (1600 nM–0.76 pM) of the recombinant fragments. Apparent dissociation constants (appK_D_) calculated for Fab, scFv HL and scFv LH were 1.0, 2.3 and 2.4 nM, respectively (Figure 4D and Table 1). An expected decrease in PSMA-binding affinity compared to that of the parent mAb (appK_D_ = 0.23 nM) most likely results from the monovalent versus divalent binding modes of fragments and full-length mAb, respectively. As affinities of both scFv variants for PSMA were virtually identical, the HL variant was directed to in vivo experiments because of approximately 10-fold higher expression yields. Affinity measurements were additionally performed using PSMA in its native environment on the surface of LNCaP cells by flow cytometry. The data were close to the result of native ELISA measurements revealing appK_D_ of Fab, scFv HL and LH to be 1.5, 1.5 and 1.3 nM, respectively (Figure 4E and Table 1).

Overall, the above in vitro experiments show that the engineered fragments recapitulate excellent characteristics of the parent 5D3 antibody well and can be thus used for further studies targeting PSMA.

### 2.5. In Vivo PSMA Imaging in an Experimental PCa Mouse Model

For in vivo experiments, engineered fragments were labeled by IRDye680RD amine reactive dye using a recently described protocol [43]. The dye-to-protein ratios of Fab, scFv HL, and mAb conjugates, as determined by ratiometric absorbance measurements, were 0.7, 1.3, and 1.8, respectively. The affinity of conjugates was evaluated by ELISA against immobilized PSMA and no noticeable decrease was observed in the affinity of conjugated proteins compared to their non-conjugated counterparts (Appendix A).

Tumors in the mouse xenograft model of PCa were formed after subcutaneous injection of PSMA-positive (PC-3 PIP) and PSMA-negative (PC-3 FLU) cells behind the front legs. Conjugated 5D3 variants were administered by intravenous injection and conjugates were localized using a fluorescence imager taking longitudinal data.

Reminiscent of full-length 5D3 mAb, conjugates of antibody fragments revealed specific localization in PSMA-positive tumors only (Figure 5). At 24 h post-injection (p.i.), conjugates of antibody fragments were cleared from the PSMA-negative tumor (PSMA−) and inner organs and a fluorescence signal was restricted to the PSMA-positive tumor only (PSMA+). Note that the signal in the gastrointestinal tract originating from chlorophyll in feed was not related to the signal of antibody molecules. Accumulation of Fab in the PSMA-positive tumor was first visible approximately 2.5 h p.i., whereas that of the scFv fragment was visible in the PSMA-positive tumor 1 h p.i. (Figure 5A). A significant difference in the signal:noise ratio of fluorescence in PSMA-positive and -negative tumors was observed 2.5 h p.i. for both fragments. However, the nonspecific signal of circulating Fab and scFv HL conjugates was cleared from the PSMA-negative tumor at 7 and 4 h p.i., respectively. This slight difference in clearance between Fab and scFv fragment was confirmed by scans of tumors and organs in situ (Appendix A) and may be related to their different molecular weights (50 and 27 kDa, respectively). The signal intensity of Fab remained unchanged until 24 h p.i., whereas the scFv conjugate was partially cleared from the PSMA-positive tumor at 24 h p.i. (Figure 5). Similarly to both antibody fragments, the 5D3 IgG was accumulated specifically in the PSMA-positive tumor without any detectable non-specific binding to the PSMA-negative tumor once the circulating conjugate was cleared. However, the conjugate of the full-length mAb penetrated tissues slowly and remained in circulation significantly longer than fragments thus revealing the specific localization of PSMA-positive tumor no earlier than 24 h post-injection. The full-length 5D3 conjugate revealed accumulation in the PSMA-positive tumor, signal:noise ratio peaking at 4–6 days post-injection. The signal persisted in the PSMA-positive tumor until 15 days p.i. (Appendix A). The IgG1 isotype control conjugated to IRDye800CW was injected together with 5D3 mAb-IRDye680RD. At 24 h p.i. the signal of both conjugates was detectable in the PSMA-positive tumor due to slow blood pool clearance. However, unbound circulating conjugates were already cleared out at day 3 p.i. and no accumulation of the isotype conjugate was detectable in the PSMA-positive tumor (Appendix A). This finding confirmed the uptake of 5D3 into the PSMA-positive tumor via its specific binding to the target PSMA antigen.

The specificity of each conjugate was further assessed in dissected tumors and organs in situ following euthanasia 24 h p.i. (Figure 5A). In line with in vivo imaging data, the specific signal of both antibody fragments was detectable in cryosections of PSMA-positive tumors, preferentially at the tumor rim and in renal proximal tubules, which are known to express high levels of PSMA and play a major role in conjugate clearance (Figure 5B). PSMA-negative tumors did not reveal any signal that would significantly differ from the background represented by a skeletal muscle. The full-length mAb conjugate signal was detectable in both tumors as well as in the kidney and a weak signal was also observed in the skeletal muscle due to blood pool content of intact immunoglobulin at 24 h p.i. as observed for in vivo images.

## 3. Discussion

In our pilot experiments, we opted for a periplasmic prokaryotic expression as this well-established system offers a simple, inexpensive, and speedy production of fully folded and functional mAb fragments [49]. Under optimized fermenter conditions, final yields of recombinant mAb fragments can be up to 5 g/L [52]. However, under our non-optimized testing conditions in shaking flasks, fragment yields were disappointing and would not allow us to carry out planned in vitro/in vivo experiments. Consequently, we turned to the insect S2 system allowing production of tens to hundreds of milligrams of secreted proteins per liter of media [53,54,55]. While bicistronic or IRES-containing plasmids can, in principle, be used for the co-expression of two genes in S2 cells [56], in the case of Fab production, we chose co-transfection of individual heavy- and light-chain encoding plasmids. In this experimental setup, one can adjust the ratio between the two plasmids to maximize yields of “complete” Fab fragments as expression levels of light and heavy chains can differ markedly [57]. In the case of 5D3 Fab, we found that the 2:1 ratio of expression plasmids resulted in the highest yield, exceeding 20 mg/L.

While recombinant Fab fragments are found almost exclusively in a monomeric form, scFvs frequently exist as a monomer/homodimer mixture. Dimerization is strongly dependent on the length of the interdomain linker and in extreme cases of short linkers (4–12 amino acids) diabodies, i.e., dimeric scFv forms, are formed preferentially [58,59]. Additionally, the length and composition of the linker has been shown to affect stability and thus functional properties of scFvs [60,61,62]. For 5D3 scFv construction, we utilized an 18-mer [Gly_4_Ser] linker, which is at the upper limit of the typically used 15–18 AA range [63], with an amino acid composition conveniently following a highly flexible sequence derived from a filamentous bacteriophage protein [64,65]. As documented by all biophysical and functional characteristics (monodispersity, temperature stability, binding affinity) the selected linker length is very well suited in the case of 5D3 scFvs.

SDS-PAGE analysis combined with mass spectrometry revealed that a significant portion of S2-expressed fragments is fused to the signal peptide upon secretion. The compromised cleavage of the signal peptide has been reported earlier and is linked to the specific features in signal sequence and timing of the cleavage (reviewed by [66]). To exclude potential negative effects of the signal sequence on the function of engineered mAb fragments, we used native ELISA to determine appK_D_ values of the Fab/scFv variant from SEC runs comprising predominantly either fusions with the signal peptide or variants with the signal peptide cleaved. All tested fractions revealed very similar appK_D_ values independently of the cleaved to uncleaved variant ratio (Appendix A). Overall, these data showed that the presence of the signal peptide does not have a significant effect on the affinity of fragments for PSMA, and the signal sequence does not therefore interfere with antibody–antigen interactions. Consequently, mixtures of uncleaved and cleaved fragments were used for all experiments without any additional purification steps. At the same time, however, heterogenous protein preparation would not meet requirements of clinical trials. Therefore, further development of 5D3 fragments should include the use of an expression system that provides a homogeneous preparation of the target protein, such as *E. coli* or mammalian-based systems used in the field. Alternatively, an enzymatic cleavage site could be inserted into constructs N-terminally to the antibody sequence to eliminate the signal peptide from the antibody fusion.

Compared to the parent intact 5D3 antibody, our engineered fragments have approximately a 5- to 10-fold lower binding strength (affinity for fragments; avidity for the whole mAb) for PSMA as determined by native ELISA. As the avidity of an antibody depends on the affinity of its binding site together with its valency, it is expected that the binding strength of monovalent mAb fragments is lower compared to bivalent parent molecules. Indeed, a decrease in affinity up to 100-fold has been documented for monovalent antibody fragments [60,61,67,68]. Importantly, our nanoDSF data show that the thermal stability of 5D3 Fab corresponds well to the stability of the original 5D3 mAb, suggesting that differences in their respective binding strengths likely stem from valency rather than lower folding stability and compactness of the Fab fragment. An optimal binding strength of antibodies, which is critical for efficient tumor uptake, is believed to be in the range of 1–10 nM [69,70,71]. Accordingly, our engineered fragments are in this desired range and can thus be used safely for further clinical development. Moreover, 5D3 fragments showed very similar characteristics to other PSMA-specific recombinant proteins. For example, a cytotoxic conjugate of scFv D7 derived from 3/F11 has appK_D_ = 6–10 nM [35]. Similarly, the RRa92 and gy1 clones of scFv were selected by phage and yeast display with an apparent affinity to recombinant PSMA of 5 nM and 1–4 nM, respectively [72,73]. scFvs derived from mouse antibodies D2B and J591 revealed appK_D_ = 9 nM and >3 nM, respectively [72,74,75]. A recombinant Fab fragment of the D2B antibody showed 9 nM affinity [23]. However, binding affinity could be enhanced by the use of a bivalent molecule. Therefore, further development of 5D3 fragments toward clinical applications could involve generation of minibody, diabody or (Fab)_2_ variants that might further improve binding characteristics.

Recombinant antibody fragments showed significantly faster clearance compared to 5D3 mAb thus confirming previously published data from a pepsin-cleaved 5D3 Fab fragment [43]. At 24 h p.i., the background fluorescence of both fragments is completely clear, whereas the mAb still shows a significant background in the whole-body scan. Consistent with data from recombinant fragments of other antibodies [67], 5D3 Fab shows the peak of specific accumulation in the PSMA-positive tumor slightly later (2.5 h p.i.) in comparison to scFv (peaking already at 1 h p.i.). Moreover, scFv starts to clear from the PSMA-positive tumor by 11 h p.i. whereas Fab is not cleared from the antigen-presenting tumor throughout the 24 h post-injection. Our data thus show standard behavior of 5D3 recombinant fragments in vivo that matches fragments of the same type derived from different anti-PSMA antibodies. For example, PSMA-directed scFv gy1 and scFv derived from the D2B Ab showed a very similar timeline of accumulation in the PSMA-positive tumor and clearance from other tissues as scFv 5D3 [72,74]. At the same time, ^111^In-D2B Fab peaks its accumulation in the PSMA-positive tumor at 48 h p.i., i.e., slightly later compared to our 5D3 fragment, suggesting the possible influence of labeling chemistry on Fab pharmacokinetics [23]. Similarly to other PSMA-specific antibody fragments [72,76], we observed signal originated from 5D3 conjugates in kidneys and liver that is connected with metabolism and clearance of recombinant proteins. Described distribution should be carefully followed particularly in future development of radionuclide or cytotoxic drug 5D3 conjugates because of potential risk of kidney toxicity [45]. The behavior of original 5D3 mAb in long time lapse was also similar to other high-affinity mouse antibodies [77], as the mAb revealed the peak of accumulation at days 4–6 p.i. and was still detectable in the antigen-presenting tumor at day 15 post-injection.

In summary, the biochemical characterization combined with data from in vivo imaging clearly show that 5D3 recombinant fragments retain an intact binding site of the original 5D3 mAb. The efficient and highly specific binding of PSMA in vivo predetermines 5D3 antibody fragments for further clinical development. This could be directed in parallel to development of other PSMA-specific tools such as bispecific molecules and fusions with toxins and immunological molecules combined alternatively with antimitotic drugs [35,78], conjugates with fluorescent dyes or radioisotopes [16,20,72], and multimeric molecules with potential to activate immune cells [31].

## 4. Materials and Methods

### 4.1. Chemicals and Reagents

All chemicals were purchased from Sigma–Aldrich (Steinheim, Germany) unless stated otherwise. Restriction enzymes, Antarctic phosphatase and ligase were purchased from New England Biolabs (Ipswich, MA, USA).

### 4.2. N-Terminal Amino Acid Sequencing

The purified antibody was mixed with the Laemmli sample buffer and individual antibody chains were separated by SDS PAGE. Proteins were transferred onto a PVDF membrane by electroblotting and visualized by staining with Coomassie Brilliant Blue G-250 (CBB). Pyroglutamate was removed from the N-terminus of the heavy chain prior SDS PAGE according to the published protocol [79]. Briefly, 5 µg of antibody was treated with 0.25 mU pyroglutamate aminopeptidase (Takara, Shiga, Japan) for 6 h at 75 °C in Reaction Buffer supplied by the producer and supplemented with 0.08% Tween-20. N-terminal amino acid sequences were determined by Edman degradation using the Procise Protein Sequencing System (494 cLC Protein Sequencer, Applied Biosystems, Foster City, CA, USA). Briefly, in each cycle, an N-terminal amino acid of an antibody chain was modified by phenylisothiocyanate, released from the polypeptide chain and converted to its stable phenylthiohydantoin derivative. The amino acid sequence was then identified by sequential analysis of amino acid derivatives using reversed-phase HPLC.

### 4.3. RNA Isolation and cDNA Synthesis

Approximately 1 × 10^7^ 5D3 hybridoma cells were homogenized in a QIAshredder spin column (Qiagen, Hilden, Germany). Total RNA was isolated by the RNeasy Mini Kit (Qiagen) according to the manufacturer’s protocol. DNase treatment was included in the isolation procedure to avoid any contamination by genomic DNA. A quantity of 1 µg of isolated RNA combined with random hexamers (30 ng/µL, Qiagen) was denatured at 65 °C for 5 min and hybridized at 4 °C for 2 min. The first cDNA strand was synthesized by mixing the hybridized template with SuperScript III reverse transcriptase (10 U/µL, Invitrogen, Thermo Fisher Scientific, Carlsbad, CA, USA), RNase inhibitor RNase OUT (1 U/µL, Invitrogen), 5 mM dithiothreitol (Fermentas, Thermo Fisher Scientific, Carlsbad, CA, USA) and 750 µM deoxynucleotide mix. Synthesis of cDNA was carried out at 50 °C for 60 min followed by inactivation at 70 °C for 15 min.

### 4.4. Antibody Cloning

Nucleotide sequences of both antibody chains encoding the 5D3 Fab fragment were amplified by PCR from the first cDNA strand using a mixture of degenerated forward primers combined with constant region-specific reverse primers. Degenerated primers were designed based on the amino acid sequence of antibody N-termini, whereas reverse primers covering C-termini of CL and CH1 domains were designed according to published antibody sequences of mouse IgG light kappa (GenBank Q58EU8) and heavy chains (GenBank U04352.1), respectively (Appendix A). Individual chains were amplified from the cDNA template (1 ng/µL) by PfuUltra II Hotstart PCR Master Mix (Agilent, Santa Clara, CA, USA) in the presence of 0.8 µM primers. Amplification was done in 35 cycles comprising denaturation (95 °C for 30 s), annealing (30 s) and extension (72 °C for 1 min) followed by final extension at 72 °C for 10 min. The annealing temperature was set to 55 and 57.5 °C for the light and heavy chain, respectively.

The pUC19 vector (Invitrogen) was linearized by the SmaI restriction enzyme, dephosphorylated by antarctic phosphatase and purified by extraction from an agarose gel using the Zymoclean Gel DNA Recovery Kit (Zymo Research, Irvine, CA, USA). Blunt-end PCR products were ligated into a linearized pUC19 vector by Blunt/TA Ligase and sequences of antibody chains were determined by Sanger sequencing (GATC Biotech, Ebersberg, Germany).

### 4.5. Mass Spectrometry Analysis of the Antibody Sequence

The amino acid sequence of the 5D3 mAb was identified by the procedure described in detail elsewhere [80]. Briefly, the protein solution was treated with dithiothreitol and iodoacetamide. After solvent removal, proteins were digested by trypsin and chymotrypsin in separate reactions. Peptides were analyzed by the UltiMate 3000 RSLCnano system (Dionex, Thermo Fisher Scientific, Rockford, IL, USA) coupled with a TripleTOF 5600 mass spectrometer with a NanoSpray III source (Sciex, Framingham, MA, USA) operated by Analyst TF 1.7 software (Sciex). The peptides were trapped, desalted and separated on an Acclaim PepMap100 column. The 125-min elution gradient at the constant flow of 300 nL/min was set to 5% of phase B (0.1% formic acid in 99.9% acetonitrile, phase A 0.1% formic acid) for first 5 min, then with gradient elution by increasing concentration of acetonitrile. Protein Pilot 4.5 (Sciex) was used for protein identification using a database consisting of proposed antibody sequences and common contaminants.

For the identification of the recombinant fragment sequence, Coomassie-stained protein bands were cut out from the gel and treated sequentially by acetonitrile, dithiothreitol and iodoacetamide. Proteins were digested by trypsin (Promega, Madison, WI, USA) in a bicarbonate buffer at 37 °C overnight. Peptides mixed with a y-α-Cyano-4-hydroxycinnamic acid matrix were subjected to MALDI-TOF analysis (Autoflex Speed MALDI TOF/TOF, Bruker Daltonik, Bremen, Germany). Data were processed by mMass software [81].

### 4.6. Construction of Recombinant Antibody Expression Vectors

To construct an expression vector for the secretion of Fab in the bacterial periplasm, variable domain sequences were amplified via PCR using conditions identical to those for the parent antibody (see above). We used sequence-specific primers (Appendix A) and an annealing temperature of 55 and 60 °C for the light- and heavy-chain variable domain, respectively. Amplified genes were ligated into a pASK85 bicistronic vector [47] containing constant domain sequences of mouse Fab tagged with a 6× Histidine anchor at the C-terminus of the heavy chain. The insertion of the light- and heavy-chain into the vector backbone was done sequentially using Eco53kI/XhoI and PstI/BstEII restriction enzyme pairs, respectively. Nucleotide strings encoding scFv variants were custom-made (Thermo Fisher Scientific). Genes were digested by the XbaI/HindIII pair and ligated into the pASK85 vector.

For S2 cell-based expression vectors, sequences from the pASK85 vector were PCR amplified using sequence-specific primers and an annealing temperature of 60 °C. PCR products were ligated via BglII/AfeI sites individually into a pMT expression vector containing a BiP signal sequence (pMT-BiP), which enabled the secretion of the target protein into the cultivation media. C-termini of genes, except the Fab light chain, were fused to the SA-strep tag. The identity of all constructed expression vectors was verified by Sanger sequencing. Primer sequences are shown in Appendix A.

### 4.7. Cell Lines

Schneider’s S2 insect cells (Invitrogen) were grown at 26 °C in Insect Xpress medium (Lonza, Basel, Switzerland) supplemented with 2 mM L-glutamine. PC-3 and DU145 cells were obtained from the American Type Culture Collection, PC-3 PIP and PC-3 FLU cells were generously provided by Dr. Warren Heston (Cleveland Clinic, Cleveland, OH, USA), and LNCaP and CW22Rv1 cells were kindly provided by Z. Hodny (IMG, Prague, Czech Republic) and R. Lapidus (University of Maryland, Baltimore, MD, USA), respectively. Human cell lines were grown in RPMI-1640 medium (Sigma-Aldrich) supplemented with 10% fetal bovine serum and 2 mM L-glutamine under a humidified 5% CO2 atmosphere at 37 °C, PC3-PIP cells were continuously selected by puromycin (20 µg/mL; InvivoGen, San Diego, CA, USA). Cells used for animal treatment were evaluated as mycoplasma-negative using MycoAlert PLUS mycoplasma detection kit (Lonza).

### 4.8. Generation of Stably Transfected Insect Cells

Schneider’s S2 cells were cultured in a medium supplemented with 10% fetal bovine serum in 24-well non-treated polystyrene plates. Cells were co-transfected by 500 ng pMT-BiP expression vector in combination with 30 ng pCoBlast selection vector (Invitrogen) using an Effectene transfection reagent (Qiagen). pMT-BiP vectors containing heavy- and light-chain of Fab were co-transfected simultaneously at a 1:2 molar ratio. Blasticidin (40 µg/mL; InvivoGen) was added to cell cultures two days after transfection and cells were maintained in the selection medium (changed every fourth day) till the selection of a stable transfected cell culture (approximately three weeks).

### 4.9. Expression and Purification of Engineered Fragments Produced in E. coli

Heterologous expression and extraction from the *E. coli* periplasm was done similarly as published earlier [82]. Briefly, JM83 cells [83] were cultured in the 2× YT medium and protein expression was induced by anhydrotetracycline (200 µg/L; stock dissolved in dimethylformamide; Acros Organics, Geel, Belgium) at 22 °C for 4 h. Proteins were extracted from the periplasm by treating cells with a periplasmic extraction buffer (500 mM sucrose, 100 mM Tris, 1 mM EDTA, pH 8) at 4 °C for 30 min. The extract was centrifuged at 5000× *g* for 15 min and the supernatant was then cleared by a second centrifugation step at 40,000× *g* at 4 °C for 30 min. Target proteins were purified by affinity chromatography using a NiNTA Superflow resin (IBA, Gottingen, Germany) equilibrated with 50 mM Tris-HCl, 150 mM NaCl, 10 mM KCl, 10% glycerol, pH 8.0. Proteins were eluted from the resin by a stepwise gradient of imidazole, elution fractions were pooled, concentrated, flash frozen in liquid nitrogen, and stored at −80 °C.

### 4.10. Expression and Purification of Engineered Fragments Produced in Insect Cells

Stable S2 transfectants were expanded to a large volume (700 mL) to the density of 3 × 10^7^ cells/mL and overexpression was induced with 0.7 mM CuSO_4_ for 7 days. The conditioned medium was harvested by centrifugation at 500× *g* followed by centrifugation of the supernatant at 10,000× *g*. The supernatant was supplemented with protease inhibitors (Roche, Basel, Switzerland) and concentrated to 30 mL using an ultrafiltration cell equipped with an Ultracel 10 kDa ultrafiltration disc (Millipore, Merck, Burlington, MA, USA). Streptavidin (8 mg/L) was added to the retentate to complex free biotin present in the culture medium. Target proteins were purified by affinity chromatography using a StrepTactin XT resin (IBA) equilibrated with 50 mM Tris-HCl, 150 mM NaCl, 10 mM KCl, 10% glycerol, pH 8.0, and eluted with 5 mM D-biotin (VWR, Radnor, PA, USA) in the equilibration buffer. Pooled elution fractions were subjected to size exclusion chromatography using a Superdex 75 column (GE Healthcare Bio-Sciences, Uppsala, Sweden) connected to an NGC Chromatography System (Bio-Rad Laboratories, Hercules, CA, USA), PBS/3% glycerol was used as the mobile phase. Fractions containing target proteins were pooled, concentrated, flash frozen in liquid nitrogen and stored at −80 °C.

### 4.11. Purification of PSMA Extracellular Domain

The expression and purification of the extracellular domain of human PSMA (residues 44–750) that comprised an N-terminal Avi-tag (Avi-PSMA) run according to protocol published earlier [54,84]. Briefly, the recombinant protein was expressed by stably transfected Schneider S2 cells and purified by the combination of affinity chromatography (Streptavidin Mutein Matrix, Roche) and size-exclusion chromatography on a Superdex 200 column (GE Healthcare Bio-Sciences). Concentrated PSMA stock was stored at −80 °C until further use.

### 4.12. SDS Polyacrylamide Electrophoresis and Western Blotting

Samples were mixed with Laemmli buffer supplemented with 100 mM dithiothreitol, heated to 95 °C and analyzed by sodium dodecyl sulfate polyacrylamide gel electrophoresis (SDS-PAGE) with CBB staining. For Western blotting, an SDS-PAGE gel was electroblotted onto a PVDF membrane in Tris-CAPS/10% methanol buffer (Bio-Rad Laboratories) using a Trans-Blot SD Semi-Dry Transfer Cell (Bio-Rad Laboratories). The PVDF membrane was blocked in 5% non-fat dry milk/PBS/0.05% Tween-20 for 45 min followed by a 45 min incubation with the Precision Protein StrepTactin-HRP (horseradish peroxidase) Conjugate (Bio-Rad Laboratories) or an anti-6× His tag antibody conjugated to HRP (both conjugates diluted 10,000× in 5% non-fat dry milk/PBS/0.05% Tween-20). Following five washes with PBS/0.05% Tween-20, the signal was developed using Luminata Forte chemiluminescence substrate (Millipore) and visualized by an ImageQuant LAS4000 Imaging System (GE Healthcare Bio-Sciences). Figures were processed using the Adobe CS4 Photoshop software (Adobe Systems, San Jose, CA, USA).

### 4.13. Indirect Immunofluorescence Microscopy

Cells were passaged on glass slides coated with poly-L-lysine and left to adhere for 24 h. Fixation was done in 3.7% formaldehyde/PBS at RT for 20 min, followed by extensive washing with PBS. Nonpermeabilized samples were directly incubated with tested antibody variants, whereas cells intended to be permeabilized were treated with 0.1% Triton X-100/PBS for 20 min and washed with PBS beforehand. Antibody variants were diluted to 14 µg/mL in PBS and incubated with slides at 4 °C overnight. Binding was detected using either anti-6× His tag antibody (2 µg/mL in PBS for 1 h; Sigma) or anti-Strep tag Ab (2 µg/mL in PBS 2 h; Immo, IBA) combined with a goat anti-mouse IgG Alexa Fluor 488 conjugate (4 µg/mL in PBS/0.05% Tween-20 1 h; Thermo Fisher Scientific). All incubation steps were followed by extensive washes with PBS/0.05% Tween-20. Finally, cells were counterstained with 6-diamidino-2-phenylindole (DAPI; 1 µg/mL) for 5 min and mounted in VectaShield (Vector Laboratories, Burlingame, CA, USA). The fluorescence signal was visualized by an Eclipse E400 fluorescence microscope (Nikon, Tokyo, Japan) equipped with a 20× and 40× magnifying dry objective. Images were taken by a ProgRes MF CCD camera (Jenoptik Optical Systems GmbH, Jena, Germany) and processed in Adobe Photoshop software.

### 4.14. ELISA

A 384-well MaxiSorp plate was coated with 20 µL/well of recombinant streptavidin (5 µg/mL in TBS) at 4 °C overnight. All further steps were carried out at RT and incubations run on a shaking platform at 700 RPM. The coating solution was discarded, and the plate blocked with 5% BSA dissolved in PBS (80 µL/well) for 2 h. Following three washes with PBS/0.05% Tween-20, Avi-PSMA (5 µg/mL) was immobilized for 1 h. After three washes, a 2-fold dilution series of antibody fragments (1600 nM–0.76 pM) and the parent 5D3 (320 nM–0.15 pM) were pipetted into the plate and incubated for 1 h. The plate was then repeatedly washed and incubated with 20 µL of a StrepMAB Classic HRP conjugate (IBA) diluted 10,000× in PBS for 45 min. TMB (3,3′,5,5′-tetramethylbenzidine) was dissolved in DMSO and further diluted in a 50 mM citrate-phosphate buffer, 0.006% H2O2, pH 5, to the working concentration of 100 µg/mL. The substrate solution (40 µL) was applied to each well and incubated for 5 min. The reaction was stopped by adding 20 µL 3 M H_2_SO_4_, and absorbance was measured at 450 nm using a ClarioStar microplate reader (BMG Labtech, Ortenberg, Germany). Data were processed in Prism 6 software (GraphPad, San Diego, CA, USA).

### 4.15. Flow Cytometry

Cells were harvested using 0.025% Trypsin/0.01% EDTA in PBS followed by a wash with 5% fetal bovine serum in PBS. Cells were then extensively washed with PBS/0.5% gelatin from cold water fish skin and incubated with antibody variants. Full-length antibodies were detected by a goat anti-mouse secondary antibody conjugated to Alexa Fluor 647 (4 µg/mL; Thermo Fisher Scientific), whereas recombinant fragments expressed by *E. coli* and insect cells were visualized by an anti-6× His tag antibody (10 µg/mL, Sigma) and an anti-Strep tag antibody (1 µg/mL, Immo, IBA), respectively, followed by incubation with a goat anti-mouse secondary antibody conjugated to Alexa Fluor 647. A three-fold dilution series of tested variants spanned the concentration of range 150 nM–0.85 pM and 500 nM–2.8 pM for mAb and fragments, respectively. All incubations were carried out in PBS supplemented with 0.5% gelatin in a total volume of 20–50 µL at 4 °C for 30 min. Incubations were followed by extensive washes using PBS/0.5% gelatin. Finally, Hoechst 33258 was added to cell suspensions to estimate cell viability. Cell samples were immediately analyzed using the LSRFortessa flow cytometer (BD Biosciences, San Jose, CA, USA). A minimum of 50,000 viable cells underwent subsequent analysis with the FlowJo software (FlowJo, LLC, Ashland, OR, USA).

### 4.16. Nanoscale Differential Scanning Fluorimetry

Standard capillaries were filled with 0.3 mg/mL protein solutions in PBS/3% glycerol. A 1.5 °C/min temperature gradient of 20–95 °C was applied to samples using a Prometheus NT.48 fluorimeter (NanoTemper Technologies, München, Germany). Melting temperatures were calculated from intrinsic protein fluorescence curves at 330 and 350 nm.

### 4.17. Fluorescence Dye Conjugation

The labeling of antibody variants was done as reported by Novakova et al. [43] with minor modifications described below. Ten micrograms of IRDye680RD-NHS or IRDye800CW-NHS esters (LI-COR Biosciences, Lincoln, NE, USA; 5 mg/mL dissolved in DMSO) were mixed with 120–160 µg proteins. Reaction mixtures were incubated at RT for 25 min and proteins were separated from the free dye using a Sephadex G-25 size-exclusion column (GE Healthcare Bio-Sciences). The presence of residual free dye in the conjugated protein solution was estimated using silica gel HLF (Analtech, Newark, DE, USA) scanned by a Pearl Impulse imager (LI-COR Biosciences) and quantified by Pearl Impulse Software (LI-COR Biosciences). The number of fluorophores conjugated per one protein molecule was calculated from protein absorbance measured at 280 nm and values of IRDye680RD and IRDye800CW absorbance measured at 672 and 780 nm, respectively. Conjugated proteins were diluted in sterile 0.9% NaCl to a final concentration of 150 µg/mL prior to injection.

### 4.18. In Vivo Imaging Using Near-Infrared Fluorescence (NIRF)

All animal studies were conducted in full compliance with the protocol approved by the Johns Hopkins University Animal Care and Use Committee. Single subcutaneous xenografts were formed in five-week old intact male athymic nude mice (Taconic Biosciences, Hudson, NY, USA) after a single injection of 3 × 10^6^ PSMA-positive PC-3 PIP cells behind a front leg and the injection of PSMA-negative PC-3 FLU cells on the opposite side [85]. Cells were inoculated subcutaneously in 100 µL of Hanks buffered saline solution. When tumors reached 4–6 mm in diameter, fluorescently labeled antibody variants (30 µg) were applied in a single dose of 200 µL into the tail vein. Mice were scanned under 2–2.5% isoflurane anesthesia in oxygen (2 L/min) at the stated time points using a Pearl Impulse imager (LI-COR Biosciences) equipped with 700 nm and 800 nm emission channels. All images were normalized to a single image in each set in Pearl Impulse software allowing for identical acquisition time and thresholding to manage comparison among time points, treatments, and animals. Immediately after the last scan, mice were euthanized using isoflurane-anesthetized cervical dislocation, the ventral side was opened, and uncovered tumors and inner organs were scanned.

### 4.19. Ex Vivo Imaging

Tumors, a kidney, and a skeletal muscle were dissected, embedded in O.C.T. compound (Sakura Finetek, Torrance, CA, USA) and frozen on dry ice immediately after euthanasia. Frozen tissues were sectioned by a Microm HM550 cryostat (Thermo Fisher Scientific) to 20 µm slides that were placed onto a positively charged glass slide and let thaw at room temperature. Dry sections were scanned by an Odyssey imager (LI-COR Biosciences) using a 700 nm emission channel and images were processed in Adobe Photoshop software.

## Figures and Tables

**Figure 1 ijms-21-06672-f001:**
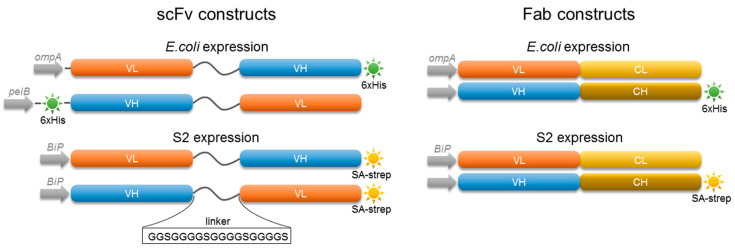
Schematic representation of 5D3 scFvs and Fab expressed by *E. coli* and insect S2 cells. Variants differ in the order of domains, the signal sequence and the type and position of an affinity tag. Signal sequences ompA, pelB, and BiP were inserted N-terminally to allow protein secretion. The SA-strep and 6× His tag positioned at either the N- or C-terminus were added for affinity purification. Variable domains of light (VL) and heavy (VH) chain were separated by an 18-mer linker identical in all scFv variants. Each Fab construct was composed of light (L) and heavy (H) chain consisting of variable (V) and constant (C) domain.

**Figure 2 ijms-21-06672-f002:**
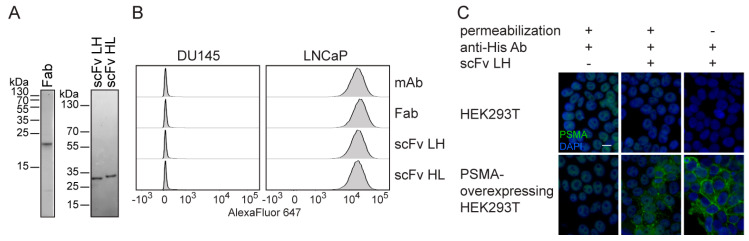
Characterization of 5D3 antibody fragments expressed in *E. coli*. (**A**) Purity of recombinant proteins was estimated from SDS-PAGE gel stained by Coomassie Brilliant Blue G-250 (CBB). (**B**) Binding of 5D3 variants on live cells was estimated by flow cytometry using anti-6× His antibody together with an anti-mouse IgG-Alexa Fluor 647 conjugate. All molecules used at saturating concentrations showed specific staining of PSMA-positive LNCaP cells with similar signal intensity. PSMA-negative DU145 cells were used as a control. (**C**) Indirect immunofluorescence microscopy using 5D3 scFv LH. PSMA-overexpressing HEK293T cells and the parent HEK293T cell line (a negative control) were fixed with/without permeabilization by Triton X-100 and incubated with 5D3 scFv LH. The fragment was then detected by anti-6× His Ab combined with anti-mouse Alexa Fluor 488 conjugate (green channel). Cell nuclei were counterstained by DAPI (blue channel). The scale bar represents 25 µm.

**Figure 3 ijms-21-06672-f003:**
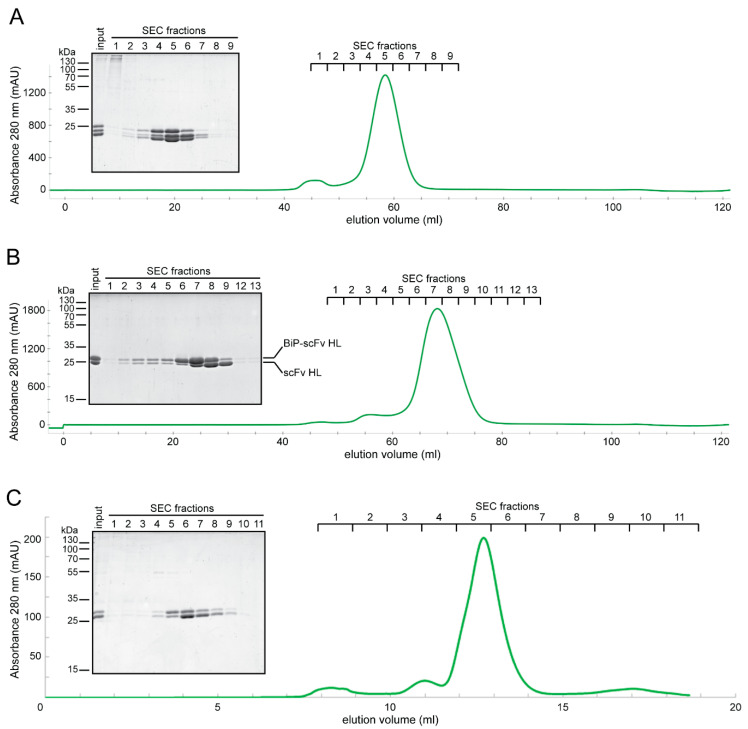
Purification of 5D3 fragments overexpressed in insect cells. Secreted variants were purified by the combination of Streptactin affinity chromatography and size exclusion chromatography (SEC). CBB-stained SDS-PAGE gels and SEC elution profiles are shown for Fab (**A**), scFv HL (**B**) and scFv LH (**C**) variants.

**Figure 4 ijms-21-06672-f004:**
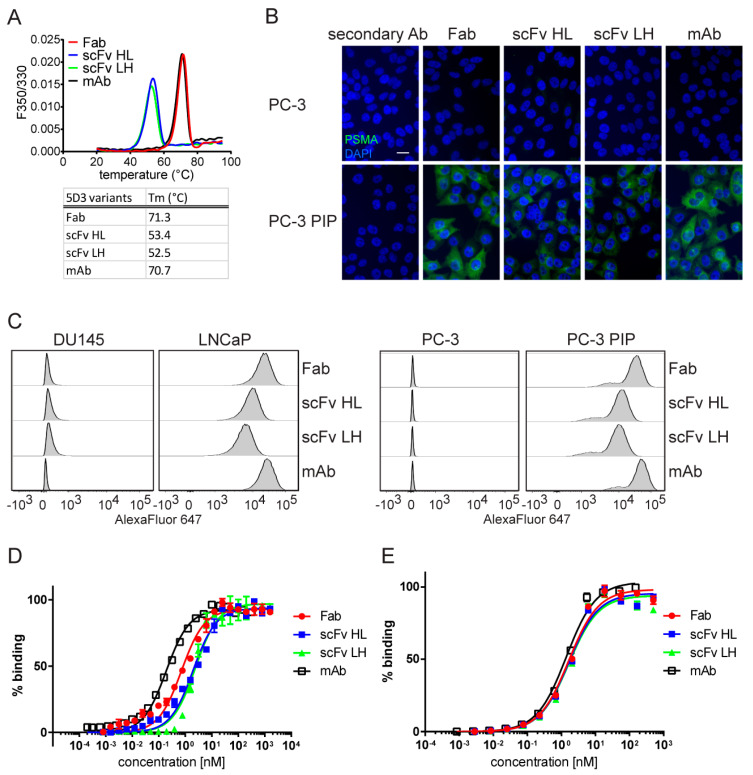
Functional characterization of S2-expressed 5D3 variants. (**A**) Thermal stability was determined using nanoDSF. Melting temperature curves and corresponding numerical values are shown. (**B**) Specificity of 5D3 variants evaluated by indirect immunofluorescence microscopy. PC-3 PIP (PSMA-positive) and parent PC-3 (PSMA-negative) cells were fixed by formaldehyde and permeabilized by Triton X-100 prior to incubation with recombinant 5D3 variants and secondary antibodies (green channel). Cell nuclei were counterstained with DAPI (blue channel). The scale bar represents 25 µm. (**C**) Flow cytometry analysis of PSMA-positive (PC-3 PIP, LNCaP) and negative (PC-3, DU145) cells stained by 100 nM purified 5D3 variants. Each gated population represents approximately 30,000 viable cells. (**D**,**E**) Determination of binding affinity of 5D3 variants by native ELISA (**D**) and flow cytometry (**E**) using purified PSMA and live LNCaP cells, respectively. PBS and DU145 cells were used as negative controls for ELISA and flow cytometry samples, respectively, and corresponding background signals were subtracted.

**Figure 5 ijms-21-06672-f005:**
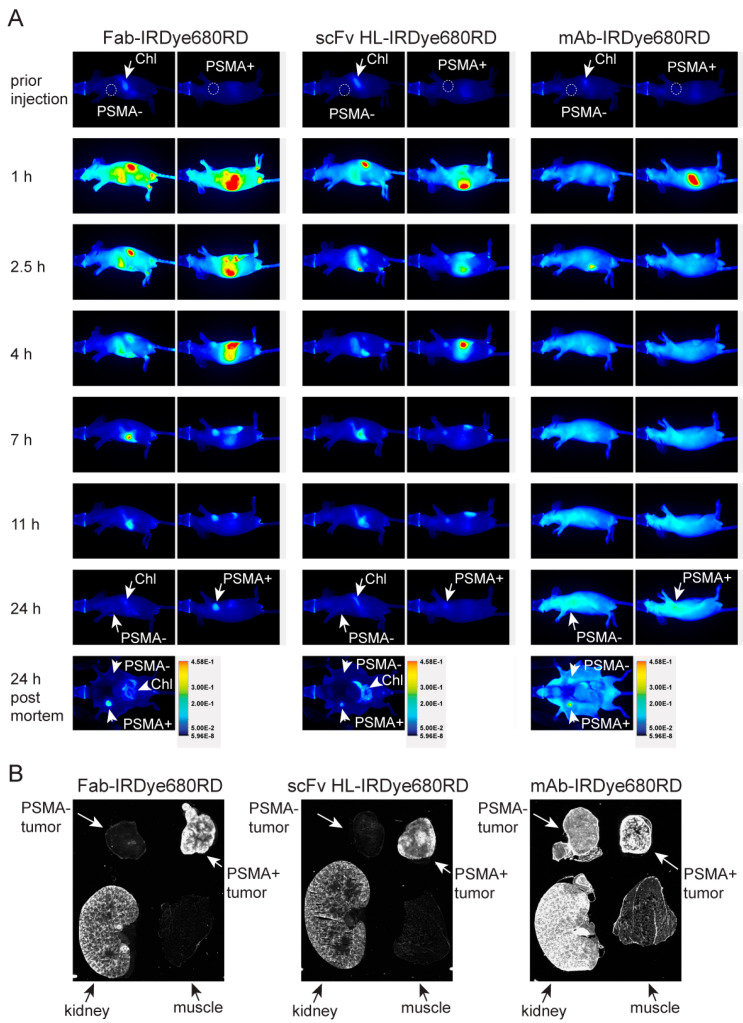
Pharmacokinetics of 5D3 variants and ex vivo NIRF imaging. (**A**) Mice bearing a PSMA-positive (PSMA+, right side) and PSMA-negative (PSMA−, left side) xenograft (location of tumor marked by dashed circle), were intravenously injected with 5D3 variants conjugated to IRDye680RD and images collected at various times post-injection with identical exposure settings. Both Fab and scFv fragments revealed specific localization to the PSMA-positive tumor already after 2.5–4 h p.i., while non-specific binding was still observed for 5D3 mAb at 24 h p.i. due to high levels of the circulating conjugate. At 24 h p.i. mice were sacrificed, and the ventral part dissected, and mice were scanned. The fluorescent signal in the gastrointestinal tract originates from chlorophyll (Chl) present in feed. Figures are representative images from triplicate runs for each 5D3 variant. (**B**) Sections of tumors, kidney and skeletal muscle (24 h p.i.) were scanned ex vivo. The specific signal for scFv and Fab was localized mainly to the rim of PSMA-positive tumors and to the kidney whereas PSMA-negative tumors and skeletal muscle did not show any significant signal. 5D3 mAb showed a strong signal in both tumors as well as kidney, and also observable signal in the PSMA-negative tumor and skeletal muscle.

**Table 1 ijms-21-06672-t001:** Apparent K_D_ of antibody fragments determined by native ELISA and flow cytometry.

	appK_D_ (nM)	
5D3 Variants	ELISA	Flow Cytometry
Fab	1.0 ± 0.7	1.5 ± 0.2
scFv HL	2.3 ± 1.7	1.5 ± 0.1
scFv LH	2.4 ± 0.1	1.3 ± 0.5
mAb	0.23 ± 0.1	0.85 ± 0.8

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
