# Peer review of "Engineered Fragments of the PSMA-Specific 5D3 Antibody and Their Functional Characterization"

_ijms, 2020, doi:10.3390/ijms21186672_

Round 1

Reviewer 1 Report

In this manuscript by Novakova et al., the authors have engineered and produced recombinant antibodies fragments (scFv and Fab) derived from 5D3 mAb, against PSMA an established biomarker for prostate cancer. 

In my opinion the manuscript can be published after minor revision:

Lane161: The authors should include the MS analysis in the manuscript or in the supplementary figures. Alternatively authors should write: data not shows.

Line192: error text, please check it.

Line 298-307: I totally agree with the authors that the signal peptide does not affect the binding, but if your goal is to go a clinical trial, i think that a mixture of uncleaved and cleaved signal pepide of antibody fragments could be problematic for regulatory authorities approval. Please comment this in the discussion section. 

Author Response

Comments from reviewer 1

In this manuscript by Novakova et al., the authors have engineered and produced recombinant antibodies fragments (scFv and Fab) derived from 5D3 mAb, against PSMA an established biomarker for prostate cancer. 

In my opinion the manuscript can be published after minor revision:

Point 1: Lane161: The authors should include the MS analysis in the manuscript or in the supplementary figures. Alternatively authors should write: data not shows.

Authors’ response: We added MS analysis of BiP fusions as a Supplementary Figure S1A.

Point 2: Line192: error text, please check it.

Authors’ response: The error on line 192 was fixed.

Point 3: Line 298-307: I totally agree with the authors that the signal peptide does not affect the binding, but if your goal is to go a clinical trial, i think that a mixture of uncleaved and cleaved signal pepide of antibody fragments could be problematic for regulatory authorities approval. Please comment this in the discussion section. 

Authors’ response: Based on reviewer’s comment we outlined options that would avoid heterogeneity of protein preparation such as switch to a different expression system or insertion of an enzymatic cleavage site prior to the antibody sequence. Explanation was inserted into the body text in Discussion section, paragraph 3.

Reviewer 2 Report

The authors reported on the development and functional characterization of recombinant single chain Fv and Fab fragments derived from the 5D3 PSMA-specific monoclonal antibody. Both In vitro assays and in vivo imaging were evaluated. The study is well designed and written. The paper is particularly interesting since it might open to development of 5D3 recombinant fragments for future clinical use.

Author Response

Comments from reviewer 2

The authors reported on the development and functional characterization of recombinant single chain Fv and Fab fragments derived from the 5D3 PSMA-specific monoclonal antibody. Both In vitro assays and in vivo imaging were evaluated. The study is well designed and written. The paper is particularly interesting since it might open to development of 5D3 recombinant fragments for future clinical use.

No detailed response required from authors.

Reviewer 3 Report

Novakova et al in this manuscript have clearly described the experimental basis for the further development of 5D3 recombinant fragments for future clinical use in the manuscript titled " Engineered fragments of the PSMA-specific 5D3 antibody and their functional characterization".

The objectives are clearly stated and the expriment's well performed in both invitro and in vivo settings. 

The authors many need to be clear why they came up with this formulation in the introduction when several others exists. 

The authors have used LNCaP, PC-3, DU-145 cells in their model system but have not shown the data in any of the CRPC counter part. They may need to discuss this issue or provide a small study in invito section.

Also, the authors have not mentioned much about the systemic toxicity of the antibody in the invivo experiments. How have they dealt with this situation.

The authors may have to provide in the discussion how this study may be scaled up and the Potential caveats that they need to overcome.

Author Response

Comments from reviewer 3

Novakova et al in this manuscript have clearly described the experimental basis for the further development of 5D3 recombinant fragments for future clinical use in the manuscript titled " Engineered fragments of the PSMA-specific 5D3 antibody and their functional characterization".

The objectives are clearly stated and the expriment's well performed in both invitro and in vivo settings. 

Point 1: The authors many need to be clear why they came up with this formulation in the introduction when several others exists. 

Authors’ response: Based on reviewer’s comment we agree that other PSMA-specific fragments have been developed and we listed them in the Introduction section. At the same time, 5D3 mAb reveals higher affinity than J591 mAb that is considered as one of leading antibodies in the field. Furthermore, as J591 patents are to expire, there might be a limited commercial potential to its further development. Since affinity is one of critical parameters that determines quality of antibody, we consider 5D3 mAb and derived fragments to be prospective competitors of the J591 family with high commercial potential. Accordingly, we mentioned superior features of 5D3 mAb in the last paragraph of the Introduction section.

Point 2: The authors have used LNCaP, PC-3, DU-145 cells in their model system but have not shown the data in any of the CRPC counter part. They may need to discuss this issue or provide a small study in invito section.

Authors’ response: We are grateful for reviewer’s suggestion about CRPC models that would enhance our knowledge about 5D3 variants. Originally, we decided to test our developed molecules on cell lines that are routinely used as a gold standard in the PCa field. Based on the reviewer’s comment we included CW22Rv1 cell line expressing low levels of endogenous PSMA. The data were inserted into Figure S1. Given the short time for revision we were not able to run other experiments with CRPC-derived cell lines as they are not currently available in our laboratory, but plan to expand the testing in the future. At the same time, we believe that using several cell lines covering the whole spectrum of PSMA expression levels is sufficient for preclinical evaluation of our fragments.

Point 3: Also, the authors have not mentioned much about the systemic toxicity of the antibody in the invivo experiments. How have they dealt with this situation.

Authors’ response: During in vivo experiments distribution of antibody fragments was documented in most of inner organs. Increased fluorescence intensity detected in liver and kidneys corresponds to other anti-PSMA antibody fragments. Based on reviewer’s suggestion we mentioned our observation in body text in Discussion section, paragraph 5. However, we did not follow systemic toxicity in detail, since we tested molecules conjugated to fluorescent dyes that are not inherently toxic. Furthermore, we do not anticipate that the toxicity profile of 5D3 variants would significantly differ from toxicity of other PSMA-specific antibodies, such as J591. Accordingly, elevated uptake of antibody fragments by kidneys represents common feature of cancer immunotherapy reagents. Since 5D3 fragments are intended for PCa imaging and therapy, conjugation of proteins with radionuclides is planned as the next step. Therefore, detailed study of toxicity would be the object of preclinical studies run on particular radioconjugates of antibody fragments. Moreover, the toxicity of 5D3 conjugate has been recently published elsewhere (Huang et al, Mol Pharmaceut, 2020, doi: 10.1021/acs.molpharmaceut.0c00457).

Point 4: The authors may have to provide in the discussion how this study may be scaled up and the Potential caveats that they need to overcome.

Authors’ response: Developed 5D3 fragments are preferentially intended for PCa therapy and imaging, since they reveal high specificity and affinity to PSMA as well as they undergo internalization triggered by antibody-antigen interaction (Huang et al, Mol Pharmaceut, 2020)). To proceed further to (pre)clinical development we plan on testing 5D3 fragments conjugated to radionuclides, e.g. 89Zr. We are aware that radionuclides could be harmful for organs responsible for conjugate clearance. Therefore, further study of radioconjugates should include detailed biodistribution documentation to follow potential toxicity in non-target tissues. Further enhancement of binding efficiency and activity of 5D3 recombinant proteins could decrease final dose and diminish potential toxicity, therefore, we are currently in the process of development of more advanced molecules optimized for in vivo applications such as minibody, diabody and immunofusion proteins (e.g. 5D3/anti-CD3). According reviewer’s recommendation we mentioned all options in body text of Discussion section, paragraph 4 and 5.